Evaluation of drought stress level in Sargent’s cherry (Prunus sargentii Rehder) using photosynthesis and chlorophyll fluorescence parameters and proline content analysis

Jin Eon Ju 1
Yoon Jun-Hyuk 1
Lee Hyeok 1
Bae Eun Ji 1
Yong Seong Hyeon 2
Choi Myung Suk mschoi@gnu.ac.kr 2 3
1 Forest Biomaterials Research Center, National institute of Forest Science , Jinju , South Korea
2 Institute of Agriculture and Life Science, Gyeongsang National University , Jinju , South Korea
3 Division of Environmental Forest Science, Gyeongsang National University , Jinju , South Korea
Montagnani Leonardo
Electronic publication date: 2023 Oct 11
Publication date: 2023
Volume: 11
Electronic Location ID: e15954
Received 2022 Oct 13; Accepted 2023 Aug 2
Copyright: ©2023 Jin et al.
Copyright year: 2023
Copyright holder: Jin et al.
License: This is an open access article distributed under the terms of the Creative Commons Attribution License, which permits unrestricted use, distribution, reproduction and adaptation in any medium and for any purpose provided that it is properly attributed. For attribution, the original author(s), title, publication source (PeerJ) and either DOI or URL of the article must be cited.
License URL: https://creativecommons.org/licenses/by/4.0/

Keywords: Sargent’s cherry, Drought stress, Photosynthesis, Chlorophyll fluorescence, Proline

Funding: The National Institute of Forest Science Research Project No. SC 0300-2019-01 This work was supported by the National Institute of Forest Science Research Project (No. SC 0300-2019-01). The funders had no role in study design, data collection and analysis, decision to publish, or preparation of the manuscript.

==============================
Sargent’s cherry trees (Prunus sargentiiRehder) are widely planted as an ornamental, climate change-sensing species. This study investigated changes in the soil moisture content, fresh weight, photosynthesis and chlorophyll fluorescence properties, and the chlorophyll and proline content of four-year-old P. sargentii seedlings after 30 days of drought stress. In the trees subjected to drought stress treatment, soil moisture content decreased, and the fresh weight of the aboveground part of the plant decreased. However, there was no significant difference in the root growth of the dried plants. Among the photosynthesis parameters, Pn MAX, E and gs showed a significant (p  <  0.001) decrease after 15 days in dry-stressed seedlings, but there was no difference between treatments in WUE until 20 days, and there was a significant (p  <  0.001) difference after 24 days. Chlorophyll fluorescence parameters, Fv/Fm, ΦPSII, Rfd, NPQ, and Pn MAX, also increased after 10 days in dry-stressed seedlings, but these changes did not reach statistical significance compared to the control treatment. These results may suggest that drought stress highly correlates with photosynthesis and chlorophyll fluorescence parameters. Chlorophyll content also significantly decreased in the seedlings under drought stress compared with the control treatment. The proline content decreased until the 10th day of drought stress treatment and increased after the 15th day, showing an increase of 10.9% on the 15th day and 57.1% on the 30th day, compared to the control treatment. These results suggest that photosynthesis, chlorophyll fluorescence parameters, and proline content can be used to evaluate drought stress in trees. The results of this study can contribute to the management of forests, such as the irrigation of trees when pore control ability and photosynthesis ability decrease.

Introduction

According to the Climate Change Commission (IPCC, 2014), the frequency of high-temperature events has increased in many countries worldwide, and the index surface temperature change is projected to exceed 2 °C by the end of the 21st century compared to 1850–1900. This scenario would cause significant environmental stress with severe consequences for the growth of crops and trees (Mittler & Blumwald, 2010; Lee et al., 2018). From January 1 to June 30, 2017, the cumulative precipitation throughout South Korea was 224.4 mm, accounting for 48.5% of the average precipitation, the worst drought since 1973 (Korea Forest Service, 2017). Because of these events, it is difficult to accurately predict and respond to water shortages. Lack of water can impair healthy plant growth and lead to plant death. Plants have a variety of survival strategies using various mechanisms to cope with water stress (Oh et al., 2005). Therefore, a complex study is required to understand the potential physiological damage to trees caused by predicted climate change and the adaptation mechanisms these trees use to combat these conditions.

In the initial response of plants to drought, factors affected by turgor pressure, such as leaf expansion and shoot elongation, are reduced, and mechanisms such as leaf detachment and stomata closure increase water conservation and water use efficiency in the plant body. Prolonged drought stress, however, causes a significant decrease in photosynthetic rate, loss of osmoregulation, and severe disturbances in significant intracellular metabolism, resulting in permanent plant damage (Taiz & Zeiger, 2006). One study found that as drought stress increased, the maximum photosynthetic rate (Pn MAX) of the Dendropanax morbiferus tree decreased (Lee, 2018), and another study found drought can affect photosynthetic capacity (Lee & Lee, 2017). According to Kim & Park (2013), dark respiration and net proton yield decreased rapidly as the period without water increased, while water utilization efficiency increased, showing decreased photosynthetic ability under poor moisture conditions. Previous studies used stomatal and non-stomatal limitations to predict the photosynthetic response to water deprivation (Drake et al., 2017; Salmon et al., 2020), while other studies found that the photosynthetic response was a valuable indicator for predicting the effect of water stress on the plant (Campos et al., 2014; Chen, Yu & Huang, 2015; Gimeno et al., 2019).

The photosynthetic ability of plants can be quantified through chlorophyll fluorescence and is used as a representative non-destructive assay to evaluate plant health. The energy absorbed by chlorophyll is: (1) used for photosynthesis, (2) emitted as a long wave by heat dissipation, and (3) the remaining dissipated energy is emitted as fluorescence (Mishra et al., 2012). Due to the competition between these three processes, chlorophyll can be used to obtain photosynthesis information (Maxwell & Johnson, 2000; Murchie & Lawson, 2013). Researchers can now measure changes quickly and easily in the structure and function of Photosystem II through the measurement of chlorophyll fluorescence in various environments to diagnose early abiotic stresses (moisture, drought, high temperature, low temperature, salt and nutrient deficiency) on plants. Although the chlorophyll fluorescence index has been widely used as a physiological indicator (Iqbal et al., 2019; Xu et al., 2020), it has not been widely tested as an indicator of drought stress or used to implement moisture management.

Several pigments are involved in photosynthesis, the most important of which is chlorophyll. Leaves have two fluorescence emission peaks, located at 685 nm of the red region (LD685) and 740 nm of the far-red region (LD740; Buschmann, 2007), which are closely related to chlorophyll content (Kalmatskaya, Karavaev & Gunar, 2016; Nyachiro et al., 2001). LD685 and LD740 are good indicators of chlorophyll and have been demonstrated to reflect photosynthetic activity (Baker, 2008; D’ambrosio, Szabo & Lichtenthaler, 1992). However, there has not yet been a comprehensive study on fluorescence kinetic parameters and the fluorescence spectrum that can be used to evaluate the response of leaves to drought stress (Magney et al., 2017).

Plants that are resistant to environmental stressors use various mechanisms to prevent damage including: organic substances in the cytoplasm, such as turgor pressure triggered by drought stress; intracellular concentration (Lichtenthaler, 1996; Bray, 1997); alleviation of osmotic stress (Kishor et al., 1995) to maintain moisture in cells; and gene expression regulation based on the specific environmental stressor. Proline plays an essential role in osmotic pressure regulation as an osmoprotectant in many plants affected by various environmental stresses, such as salinity and drought stress (Giri, 2011; Semida et al., 2015; Arteaga et al., 2020). Energy and amino nitrogen storage have been reported to play an important role in the rapid restoration of cellular homeostasis and recovery after drought stress (Verbruggen et al., 1996), and proline accumulation may be part of the stress signal influencing these adaptive responses (Maggio et al., 2002).

The Sargent’s cherry tree (Prunus sargentii Rehder) is a broad-leafed, deciduous tree that belongs to the Rosaceae family and is native to Korea (Fig. 1A). It has strong cold resistance, so it can grow anywhere in the country, but grows particularly well on the seaside. As a shade-intolerant shade tree, P. sargentii thrives in flat, fertile soil with high humidity, grows very quickly and has strong resistance to air pollution (Cho & Choi, 1992). Considering the growth characteristics of this species, the Korean Forest Service has also recommended P. sargentii for reforestation. According to statistical data from the Korea Forest Service (2020), cherry trees are currently the most planted species (1,546,857 trees), accounting for 17.9% of trees planted on Korean streets in 2020. And in the National Preferred Tree Survey (Korea Forest Service, 2022), Cherry trees (16.2%) were selected as the 3rd favorite tree by Koreans, following pine trees (39.3%) and maple trees (16.6%).

Figure 1 Sargent’s cherrytree (Prunus sargentii Rehder) (A) and overview of the drying treatment in this study (B).

Due to climate change, Sargent’s cherry trees have recently started to wither in street planting sites. It is difficult to plant and manage roadside trees due to the significant lack of abiotic and physiological data such as the amount of moisture and light needed by wild cherry trees planted in these conditions. Understanding the physiological responses of different cherry tree species to drought stress would be helpful for selecting and managing cherry trees in Korean cities. This study investigated the physiological mechanisms used by Sargent’s cherry trees against drought stress. The following hypotheses were tested: (1) soil moisture content is significantly correlated with growth, photosynthesis, and chlorophyll fluorescence parameters of grafted Sargent’s cherry trees; (2) Sargent’s cherry trees increase water utilization efficiency while maintaining photosynthetic efficiency in dry conditions; and (3) the degree of drought resistance of Sargent’s cherry trees could be identified by analyzing soil moisture content, chlorophyll fluorescence response, and proline content. This study sought to identify the optimal environmental moisture conditions of Sargent’s cherry trees and the drought resistance mechanisms this species uses by examining various physiological responses to continuous drought stress.

Materials & Methods

Planting materials, experimental design, and environmental variables

The four-year-old Sargent’s cherry (Prunus sargentii Rehder) tree used in the experiment is a seedling grafted with an annual branch collectedin January 2017 from the Sargent’s cherry Tree Genetic Resource Conservation Center (E126°56′03″, N33°31′06″) of the Warm Temperate and Subtropical Forest Research Center of the National Institute of Forest Science. Grafted seedlings were grown in a greenhouse (E128°10′08″, N35°16′33″) in the Forest Biomaterials Research Institute of the National Institute of Forest Science, and 100 grafted seedlings were transplanted into a 40 L air pot in March 2021. The Soil used for transplantation was mixed with Masato and bed soil in a ratio of 1:1, and grafted seedlings were used in the experiment after being acclimatized in a greenhouse for 5 months before drought stress treatment.

Drought stress was induced through artificial water treatment for about 1 month from August 1, to August 31, 2021. Among the 100 individual transplanted trees, 66 individual trees (root diameter 13.0 ± 2.6 cm, height 2.0 ± 0.4 m) were divided into control trees (10) and treatment trees (56: 8 individuals; 7 times measurement). Direct irrigation was conducted on the control trees to maintain the soil moisture content at 15.0 ± 0.5% until the end of the study (Fig. 1).

After irrigation stopped, a temperature and humidity measuring device (HOBO H08-004-02, ONSET, USE) was installed 1m above the ground to measure environmental factors in the greenhouse during the period of the experiment. A Photon Systems Instrument (Drasov, Czech) was used every day from 13:00 to 14:00. During the experiment, the average temperature was 24.2 ± 5.7 °C, the highest temperature was 37.6 °C, the lowest temperature was 12.5 °C, and the average daily temperature difference during the experiment was 15.3 °C (18.1∼28.4 °C), which is a relatively large difference (Fig. 2A). The average relative humidity was set to 68.6 ± 18.9% (Fig. 2B). The average solar radiation was set to 468.1 W mm−2.

Figure 2 Changes of mean air temperature (A), solar radiation (B) and relative humidity (C) green house on during the experimental period.

Measurement of growth parameters and soil water content

To compare the effect of drought stress on growth, three specimens were collected at intervals of five days, divided into aboveground parts (stems, leaves) and underground parts (roots), and the fresh weight of each part was measured. After fresh weight was measured, these parts were washed thoroughly with tap water, and dried in a dry oven at 70 °C for 48 h, and then the dry weight of each part was measured. Soil moisture content was measured 5 times every 20 min at a depth of 10 cm on the soil surface using a smart soil moisture sensor (S-SMD-M005, Onset, Buzzards Bay, MA, USA).

Analysis of photosynthetic measurements

Photosynthesis was measured in healthy leaves per unit leaf area using a portable photosynthesis system (Portable Photosynthesis system, Li-6400, Li-Cor Inc., Lincoln, NE, USA) from 09:00 to 15:00 on a sunny day, when photosynthesis is active. The following photosynthetic measurements were taken at five-day intervals, with 15 repetitions per object (5 leaves × 3 individual trees), measured seven times: maximum photosynthesis rate (Pn MAX), stomatal transpiration rate (E), stomatal conductance (gs), water use efficiency (WUE).

Photosynthetic Photon Flux Density (PPFD) controlled the light intensity using an LED light source attached to a portable photosynthetic measuring device in eight steps (0, 100, 200, 400, 800, 1,000, 1,400, and 1,800 µmol m−2 s−1).

The air flow into the chamber was kept at 500 µ s−1 and the temperature was 20 ± 2 °C during all photosynthetic measurements. All measured data were automatically saved in the Date Logger, and the maximum photosynthetic rate, stomatal transpiration rate, and stomatal conductivity per unit leaf surface area were automatically calculated using the formulas of Von Caemmerer & Farquhar (1981) and expressed as the value obtained by dividing the transpiration rate, µmolCO2 mmol H2O−1.

Analysis of chlorophyll fluorescence

A total of 105 chlorophyll fluorescence measurements were taken: 15 repetitions each (five leaves of three individuals) every five days for 30 days from the day watering stopped. Measurements were taken between 13:00 to 14:00. For the first 10 days of drought stress treatment, the 13th to 15th leaves from the growing point were measured. After the 15th day of treatment, the 7th to 10th leaves from the growing point were measured. The same leaves were used for both the photosynthesis measurements and the chlorophyll fluorescence measurements. The leaf clip was bitten on the plant leaf before measurement and irradiated after 20 min of dark treatment. Fv/Fm, ΦPSII,, RFd, and NPQ were measured using a quenching kinetics analysis after 20 min of dark treatment in a chlorophyll fluorescence analyzer chamber using a Handy Cam (FlorCam, CZ; Barbagallo et al., 2003; Genty et al., 1990). Continuous, actinic light (red LED) was used at a moderate light amount of 200 µmol m−2 s−1 and a saturating light amount of 1,250 µmol m−2 s−1 to induce chlorophyll fluorescence for the measurements. The measured data were analyzed using the methods presented by Gorbe & Calatayud (2012). PIABS was calculated using a chlorophyll fluorescence meter (FP-100, Photon System Instruments, Czech Republic) according to the JIP-TEST method (Stirbet & Govindjee, 2011; Table 1).

Table 1 Chlorophyll fluorescence parameters used in this study.

Parameter	Formula	Description	
Fv/Fm	(Fm- Fo)/Fm	Maximum quantum yield of PSII photochemistry measured in the dark-adapted state	
ΦPSII	(F’m- Fs)/F’m	Effective quantum yield of photochemical energy conversion in PSII	
Rfd	(Fm−Fs)/Fs	Ratio of fluorescence decline	
NPQ	(Fm -F’m)/F’m	Non-photochemical quenching of maximum fluorescence	
PIABS	RCABS⋅ΦPo1−ΦPo⋅Ψo1−Ψo	Performance index on absorption basis	

Analysis of chlorophyll content

Chlorophyll content measurement was compared and analyzed after collecting leaves every five days for 30 days (seven times in total) after watering ceased. Chlorophyll was extracted from leaves using dimethyl sulfoxide (DMSO) as an extraction solvent according to the methods outlined by Hiscox & Israelstam (1978). The extract was obtained by measuring the absorbance at wavelengths of 663 nm and 645 nm using a UV-Vis spectrophotometer (Nicolet Evolution 100, Thermo Electrom Co., USA), and chlorophyll a and b content were obtained by the following formula (Arnon, 1949; Mackinney, 1941).

Analysis of proline content

Proline analysis was performed according to the methods outlined by Bates, Waldren & Teare (1973). The leaves were collected before the drying treatment every five days for 30 days (seven times in total) after watering ceased. After collecting 0.1 g (15 total repetitions) of each leaf, 10 mL of a sulfosalicylic acid solution (3%, w/v) was added, followed by mortar grinding. The grinding solution was filtered with two layers of filter paper (Whatman No. 42). After adding 1 mL of glacial acetic acid and 1 mL of ninhydrin reagent to 1 mL of the filtrate, the test tube was capped, reacted in boiling water (100 °C) for one hour, and then stored at room temperature (21.0 °C) for five minutes. Then, two mL of toluene was added, stirred for 20 s, and then the supernatant was taken, and the wavelength was measured at 520 nm using a UV spectrophotometer (X-ma 2000, Human Crop.). Quantitation was calculated according to a calibration curve prepared using proline (Sigma-Aldrich Co., St. Louis, MO, USA) as the standard material and expressed as µmol proline/g FW.

Statistical analysis

The homogeneity of data variance was tested using Levene’s test. Data on physiological indicators were analyzed using SPSS software (ver. 27.0; SPSS Inc., Chicago, IL, USA) by one-way ANOVA, which takes the elapsed time after a single treatment as a factor. Duncan’s multiple range test determined the difference between averages at the 5% significance level (DMRT, p < 0.05). Before performing the analysis of variance, the data sets were checked for homogeneity of error variances using the Shapiro–Wilk test in SPSS software to ensure that the homogeneity assumption was not violated. In addition, Pearson’s correlation analysis by the R statistical package (R-x64-4.0.4) was performed on the correlation between each physiological indicator of drought stress.

Results

Effect of drought stress on plant growth and changes in soil moisture content

As expected, the decrease in soil moisture content was significantly higher in the drought stress treatments than in the control (Fig. 3). Immediately after irrigation stopped, soil moisture content was 20.1% in both the control and treatment groups. On the 2nd-10th day after watering ceased, soil moisture content ranged from 22.7∼18.4%; it ranged from 9.6∼5.4% on the 15th-19th day after watering ceased, and fell to 1% or less after the 20th day of drought treatment. The aboveground soil moisture increased by 4.3% in the control and decreased by 17.9% in the drought stress treatment group, and the underground soil moisture increased by 3.5% in the control group and decreased by 7.2% in the treatment group, when overall soil moisture content in this group was less than 10%.

Figure 3 Changes in visual appearance of Prunus sargentii seedlings (A) soil water content (B) shoot (C) and root (D) fresh weight drought stress conditions during treatment time.

Different letters indicate a significant difference difference at p < 0.05 by Duncan’s multiple range test.

Effect of drought stress on leaf photosynthetic traits

The maximum photosynthetic rate, stomatal transpiration rate, stomatal conductivity, and water utilization efficiency measured in P. sargentii leaves showed significant differences between drought stress treatment and control as the experimental period increased (p < 0.05; Fig. 4). The maximum photosynthetic rate in the drought stress treatments showed a significant 29.8% decrease to 7.22 ± 0.66 µmol CO2 m−2 s−1 on the 15th day of no watering, compared to before drought treatment, and fell to 2.15 ± 0.79 µmol CO2 m−2 s−1 on the 30th day, an 80.1% reduction (Figs. 4A, 4B). This sharp decrease in the maximum photosynthetic rate after 15 days of drought stress coincided with a fall in soil moisture content from 9.6 to 5.4%.

Figure 4 Variations of photosynthetic characteristics under control and drought stress of Prunus sargentii.

(A, B) Maximum photosynthesis rate. (C, D) Stomatal transpiration rate. (E, F) Stomatal conductance. (G, H) WUE (water use efficiency).

There was no significant difference in stomatal conductance between drought stress treatment and control throughout the study period (Figs. 4C–4F). There was, however, a significant difference in stomatal transpiration rate between treatment (1.35 ± 0.04 mol CO2 m−2 s−1), and control trees (0.97 ± 0.02 mol CO2 m−2 s−1) after 15 days of drought treatment, with a 33% decrease in the treatment group from the start of the study. This decrease coincided with the significant drop in maximum photosynthetic rate, and the soil moisture content falling below 10%. Pore conductivity also showed a significant difference between groups after 15 days of drought treatment, with the pore conductivity of the treatment group 77.7% lower than the control group (0.16 ± 0.01 mol CO2 m−2 s−1 vs. 0.04 ± 0.00 mol CO2 m−2 s−1).

Compared to the control, water utilization efficiency temporarily increased after 10 days of drought stress treatment, then decreased from the 15th day when the maximum photosynthetic rate decreased (Figs. 4G, 4H).

Effect of drought stress on leaf chlorophyll fluorescence response

The four measured indices of the chlorophyll fluorescence response all decreased after the 15th day of drought stress treatment, showing a significant difference between the groups. After the 20th day, these four measurements sharply decreased (Figs. 5A, 5B). Fv/Fm, which shows the maximum quantum yield of photosystem II in the dark adaptation state, was 0.84 ± 0.02 in the control group before treatment and 0.80  ± 0.01 on the first day of drought stress treatment. After 15 days, Fv/Fm was 0.82 ± 0.02 in the control group and 0.57 ± 0.04 in the drought stress treatment group, a 0.2% and 2.88% decrease, respectively. After 30 days of drying treatment, Fv/Fm decreased 4.8% in the control group and 43.4% in the treatment group, to 0.78 ± 0.03 and 0.45 ± 0.02, respectively.

Figure 5 Variations of photosynthetic characteristic in control and drought stress.

(A, B) Fv/Fm, (C, D) ΦPSII, (E, F) Rfd, (g, h) NPQ, (I, J) PIABS. In the box plot, the points and short error bars represent the mean (±SE) of n = 21 per treatment, and the line and long error bars represent the median line and 95% CI, respectively. In the line chart, the points and error bars reflect the mean (±SE) of three replicates per treatment per date. The blue and red indicates the control and drought treatment, respectively.

Chlorophyll fluorescence (Rfd) was 5.16 ± 0.32 in control and 5.21 ± 0.09 in the treatment group at the beginning of the study, prior to drought stress treatment. After 15 days of drought stress treatment, Rfd was 5.20 ± 0.09 in the control group and 2.35 ± 0.15 in the treatment group, 54.9% lower than the control group. After 30 days of drought stress treatment, Rfd in the control group increased by 2.6% to 5.30 ± 0.11, and decreased by 83% in the treatment group to 0.89 ± 0.02, indicating Rfd is sensitive to drought stress (Figs. 5E, 5F).

Non-optical fluorescence extinction (NPQ) also showed a significant difference between groups after 15 days of drought stress treatment, with NPQ in the control group decreasing by 1.0% to 3.02 ± 0.65, and NPQ in the drought treatment group decreasing 43.3% to 1.02 ± 0.05. After 30 days of drought treatment, NPQ in the control group increased by 6.7% to 3.31 ± 0.11, and NPQ in the treatment group decreased by 83.2% to 0.62 ± 0.04 (Figs. 5G, 5H).

PIABS also showed a significant decrease in the treatment group compared to the control group, with PIABS in the treatment group decreasing 32.4% in the first 15 days of drought stress treatment to 5.48 ± 1.07, followed by a decrease of 82.1% after 30 days (Figs. 5I, 5J).

Overall, the chlorophyll fluorescence response significantly decreased after drought stress treatment compared to control. After the 15th day of drought stress treatment, the energy captured for use in the photochemical process decreased. The energy not used for electron transfer increased, indicating reduced activity.

Effect of drought stress on leaf chlorophyll response

Chlorophyll content showed a significant difference between groups (p < 0.05). Chlorophyll a and b content showed similar values until the 10th day of drought stress treatment (Fig. 6), but these values were significantly lower in the treatment group after 15 days of treatment, compared to the control group. Total chlorophyll content decreased by 13.8% to 4.98 ± 0.21 mg g−1 in the control group in the first fifteen days of the study, while the total chlorophyll content in the treatment group decreased by 42.1% to 3.80 ± 0.11 mg g−1, and further decreased by 77.8% to 1.46 ± 0.25 mg g−1 after 30 days of drought stress treatment.

Figure 6 Variations of photosynthetic characteristic in control and drought stress.

(A, B) Chl. a, (C, D) Chl. b, (E, F) Total Chl., (G, H) Chl. a/b. In the box plot, the points and short error bars represent the mean (±SE) of n = 21 per treatment, and the line and long error bars represent the median line and 95% CI, respectively. In the line chart, the points and error bars reflect the mean (±SE) of three replicates per treatment per date. The blue and red indicates the control and drought treatment, respectively.

Effect of drought stress on leaf proline response

The proline content in the control group was 2.24 mg at the start of the study and 1.67 mg after 30 days (Figs. 7A, 7B), but this difference did not reach statistical significance. However, drought stress treatment caused significant decreases in proline content. Proline content did not decrease significantly in the first 10 days of drought stress treatment, from 1.52 ± 0.02 to 1.51 ± 0.06 mg, but decreased 11.6% to 1.35 ± 0.08 mg on the 15th day of treatment, and by 61.1% to 3.90 ± 0.18 mg after 30 days of drought stress treatment. The most significant decreases in proline content occurred at the same time the fresh weight, photosynthesis and chlorophyll fluorescence values decreased, and when the soil moisture content fell below 10%.

Figure 7 Variations of proline in control and drought stress. (A, B) Proline.

In the box plot, the points and short error bars represent the mean (±SE) of n = 21 per treatment, and the line and long error bars represent the median line and 95% CI, respectively. In the line chart, the points and error bars reflect the mean (±SE) of three replicates per treatment per date. The blue and red indicates the control and drought treatment, respectively.

Correlation among factors

A correlation analysis was performed between photosynthesis, chlorophyll fluorescence, chlorophyll and proline content activity of P. sargentii trees (Fig. 8 & Table 2). PnMAX (r = 0.98∗∗∗), PIABS (r = 0.96∗∗∗), Rfd (r = 0.93 ∗∗∗), and gs (r = 0.90 ∗∗∗) were all positively correlated with E, and proline content (r = −0.74∗) was negatively correlated with E. Proline content was negatively correlated with all other parameters except for WUE (r = 0.69), with which it was positively correlated. The following photosynthetic properties and chlorophyll fluorescence parameters were positively correlated: Fv/Fm to gs (r = 0.78 ∗), ΦPSII to gs (r = 0.97∗∗∗) ΦPSII to E (r = 0.91∗∗∗), ΦPSII to Pn MAX (r = 0.89 ∗∗), Rfd to gs (r = 0.98∗∗∗), Rfd to E (r = 0.94∗∗∗), Rfd to Pn MAX (r = 0.93∗∗∗), NPQ to gs (r = 0.95∗∗∗), NPQ to E (r = 0.87∗), NPQ to Pn MAX (r = 0.88∗∗), PIABS to gs (r = 0.86 ∗∗), PIABS to E (r = 0.97∗∗∗), and PIABS to Pn MAX (r = 0.96∗∗∗). There was a significant positive correlation between chlorophyll content and E, Fv/Fm, ΦPSII, Rfd, and a significant negative correlation between chlorophyll content and proline content.

Figure 8 Correlation analysis for photosynthesis, chlorophyll fluorescence parameters, chlorophyll, and proline contents in P. sargentii seedlings, regardless of treatment length or drought stress.

Blue and red boxes represent positive and negative correlation, respectively. Color intensities are proportional to the correlation coefficients, as shown in the legend to the right.

Table 2 Summary of analysis of variance for photosynthesis characteristics, chlorophyll fluorescence, chlorophyll, proline assay of Prunus sargentii at the two water levels (control, drought stress) and drought treatment times.

Parameters	Water level (W)	Treatment time (T)	W × T	
	F-value	Significance	F-value	Significance	F-value	Significance	
Pn max	344.0	***	65.7	***	57.3	***	
E	305.5	***	51.7	***	29.1	***	
gs	963.6	***	213.1	***	83.7	***	
WUE	29.8	***	8.1	***	6.6	***	
Fv/Fm	14.8	**	14.2	**	4.8	**	
ΦPSII	635.7	***	87.1	***	92.8	***	
Rfd	780.9	***	51.6	***	88.4	***	
NPQ	258.8	***	14.3	***	23.2	***	
PIABS	202.9	***	27.0	***	13.4	***	
Chl. a	12.2	**	6.2	*	2.4	*	
Chl. b	10.8	**	12.1	*	2.6	*	
Total Chl.	13.4	**	7.4	*	1.8	*	
Chl. a/b	6.7	*	13.9	*	2.9	*	
Proline	136.7	***	132.1	***	97.9	***	
Notes.

RMANOVA was used estimate the effect of treatment: *, **, and *** indicate significance at p < 0.05, p < 0.01, and p < 0.001, respectively.

NS non-significant

Discussion

Changes in plant growth and soil moisture content after drought stress treatment

Moisture and temperature affect the growth and physiological characteristics of trees (Wu, Jansson & Kolari, 2011; Rustad et al., 2001). Drought stress is a significant limiting factor in the initial growth and establishment stages of plants, affecting both cell length growth and hypertrophy (Kusaka, Ohta & Fujimura, 2005; Shao et al., 2008). In general, when plants are under drought stress, they reduce the ratio of aboveground to underground parts and develop deeper roots to reduce water consumption and enhance water uptake (Pallardy & Rhoads, 1993). In this study, as the soil moisture content decreased, the aboveground fresh weight decreased compared to the control group. However, the underground fresh weight was higher than the control treatment until the 20th day of drought stress treatment, when the underground fresh weight started to decrease, but these differences did not reach statistical significance (p > 0.99) between the two groups throughout the study period. Zang et al. (2014) divided beech trees into a normal drying zone and a strong drying zone, and found that root production increased in the normal drying zone, but root production decreased in the strong drying zone, and the ratio between root to shoot biomass increased. In the present study of Sargent’s cherry trees, the aboveground fresh weight did not change significantly until the soil moisture content fell to around 5.0%, after 25 days of drought stress treatment, indicating that prolonged drought stress impacted both the aboveground and underground parts of the tree. Previous studies have shown that poor root respiration in plant growth affects the synthesis of new plant tissues and the preservation of living tissues (Ryan & Law, 2005; Lee et al., 2012), and a decrease in root respiration results in the loss of anabolic capacity. A previous study reported that root growth was restricted as it led to a decrease in root respiration (Bengough et al., 2006). This study found that more assimilation materials were directed to the underground part of the plant rather than the aboveground part in response to short-term drought stress, however, more research is needed on the mechanisms used in response to long-term drought stress.

Response of leaf photosynthetic traits to drought stress

Drought stress induces plants to close their stomata, reducing the CO2 concentration in the mesophyll, thereby directly inhibiting photosynthesis or inhibiting carbon metabolism, resulting in reduced photosynthesis (Gimenez, Mitchell & Lawlor, 1992; Cornic, 2000). In this study, the photosynthetic rate, transpiration rate, and stomatal conductance of the trees subjected to drought stress decreased compared to the control trees (Fig. 4). A decrease in these photosynthetic characteristics due to drought stress can decrease plant growth. Many research studies have been reported on the effect of drought stress on photosynthesis, and the decrease seen in photosynthetic efficiency is known to be due to various causes (Chaves & Oliveira, 2004). Abscisic acid (ABA) is synthesized when plant roots sense water stress. ABA moves through the xylem, induces various actions such as stomatal control (Zhang & Davies, 1990), and activates defense mechanisms against stress. This study confirmed that the resistance to drought stress was increased by quickly controlling the opening and closing reaction of the stomata through E measurements. Water utilization efficiency is closely related to plant growth, and plants close their stomata to increase their efficiency in a water-poor environment, reducing the transpiration rate more than photosynthesis. However, this efficient increase in power negatively correlates with plant growth (Richards & Condon, 1993). This study found that water utilization efficiency increased when the transpiration rate was reduced by closing the stomata. However, plant growth deteriorated due to the decrease in photosynthetic rate.

Response of leaf chlorophyll fluorescence to drought stress

Drought is an abiotic stressor that affects photosynthesis in the short and long term due to the stomatal closure in plants and the inactivation of RuBisCo (Gorbe & Calatayud, 2012). Fv/Fm is a representative chlorophyll fluorescence index that can evaluate the photosynthetic level of plants during dark adaptation and is used to detect various abiotic and biotic stresses (Rungrat et al., 2016). In this study, the Fv/Fm value decreased after 15 days of drought stress treatment (Fig. 5A). It is presumed that drought stress inhibited the photochemical activity of photosystem II and reduced the Fv/Fm of the leaves. PSII can also be damaged under drought stress, inhibiting the primary reactions of photosynthesis (Lichtenthaler & Rinderle, 1988). Fluorescence parameters in leaves are known to be altered in two ways under stress conditions: minimal fluorescence (Fo) increases due to obstruction of electron flow through PSII, and plastoquinone receptor (QA-) cannot be fully oxidized during stress. The decrease in Fm during drought stress may also be influenced by the reduced activity of water lyase complexes and accompanying cyclic electron transport in or around PSII (Porcar-Castell et al., 2014).

In PSII, the maximum fluorescence value (Fm_LSS) is measured by irradiating saturated light while the plant is photosynthesizing. In this state, when actinic light (light that causes photosynthesis) is continuously illuminated, fluorescence decreases and reaches a steady state consisting of Ft_LSS representing the photochemical energy conversion efficiency of photosystem II (Schreiber & Bilger, 1993; Stępień & Kłbus, 2006; Krause & Weis, 1991; Baker, 2008; Boughalleb, Denden & Tiba, 2009). After 15 days of drought stress treatment, the PSII value decreased by 56.0%, indicating it was more sensitive to drought stress than Fv/Fm (Figs. 5C, 5D). There was also a significant decrease in PSII after 15 days of drought stress treatment, indicating that CO2 supply was reduced due to stomatal closure (Zhou et al., 2017). Chlorophyll fluorescence reduction (Rfd) reflects photosynthetic performance. When measured under saturated light, Rfd correlates with CO2 fixation rate (Lichtenthaler et al., 2005) and decreases as drought stress increases (Méthy, Olioso & Trabaud, 1994). In this study, Rfd significantly decreased after 15 days of drought stress treatment, when the photosynthetic rate also began to decrease significantly. Photosynthetic efficiency is reduced when the water potential of the leaves and the photosynthetic rate are also reduced (Lawlor & Cornic, 2002; Chaves & Oliveira, 2004).

NPQ, which refers to the thermal loss of energy in the photosynthetic mechanism during photochemical energy conversion, is known to increase under stress conditions (Genty et al., 1990), but in this study, photosynthesis and transpiration rates decreased in the first 15 days of drought stress treatment before decreasing even more sharply (Figs. 5G, 5H). This was consistent with previous studies that showed that damage to photosynthetic pigments reduced chlorophyll fluorescence and decreased NPQ (Shin et al., 2021; Kim et al., 2020). However, since NPQ is related to the thermal dissipation of leaves, a comprehensive study considering leaf temperature is necessary to understand this relationship in conditions of drought stress.

PIABS, which represents the photochemical performance index of photosystem II or the vitality level of the plants, significantly decreased as drought stress time increased, falling 82.1% (Figs. 5I, 5J) after 30 days of drought stress treatment compared to the beginning of the study. This suggests that when the soil moisture content of Sargent’s cherry trees is less than 5.0%, the energy captured for use in the photochemical process decreases and the energy not used for electron transfer increases, resulting in a decrease in photosystem II activity. PIABS represents the energy conservation efficiency in electron carrier reduction using absorbed light energy (Holland, Koller & Brüggemann, 2014), and is used to evaluate the degree of stress and photosynthetic capacity of plants (Van Heerden, Swanepoel & Krüger, 2007), with lower PIABS levels indicating higher levels of stress (Wang et al., 2012). PIABS results in this study indicate that the soil moisture content of Sargent’s cherry trees should be kept at 5.0% or more for stable growth.

Response of leaf chlorophyll traits to drought stress

Proline is a crucial osmotic regulator and free radical scavenger that can alleviate stress damage by reducing water potential (Hayat et al., 2012). We found that proline content gradually increased during drought stress treatment, with a significant increase in proline content when the soil moisture content was less than 5.0% (Fig. 7). This decrease in proline content is thought to be related to the osmotic adjustment mechanism (Xiao, Xu & Yang, 2008) that protects plants from dehydration due to drought stress and lowers osmotic potential. Also, as proline content increased, photosynthetic efficiency significantly decreased, likely due to the decrease in stomatal conductance that increases the accumulation of ABA content. Several studies have shown that proline accumulates in dehydrated conditions and is rapidly lost when dehydration conditions are relieved (Blum & Ebercon, 1976; Singh, Aspinall & Paleg, 1973; Stewart, 1972). When osmotic stress is removed, proline is oxidized to Δ1-pyrroline-5-carboxylate (P5C) by proline dehydrogenase, also known as proline oxidase, the first enzyme in the proline degradation pathway. P5C is then converted back to glutamate by the enzyme P5C dehydrogenase (Hare, Cress & Van Staden, 1998).

Correlation among factors

Photosynthesis and chlorophyll fluorescence were positively correlated, with both factors significantly decreasing with increased drought stress. A previous study showed that reduced chlorophyll fluorescence parameters following drought stress impaired photosynthetic electron transport (Zhuang et al., 2020). In this study, Pn MAX showed the highest positive correlation with PIABS (r = 0.96***) and Rfd (r = 0.93***). Drought stress damages the reaction center of PSII and inhibits the electron transfer process of photosynthesis, reducing the photosystem II efficiency of light energy conversion (Brestic et al., 1995; Cornic & Fresneau, 2002; Longenberger et al., 2009). Drought stress also alters the structure of the leaf chloroplast layer and reduces chlorophyll content (Batra, Sharma & Kumari, 2014). Chlorophyll content showed the highest positive correlation with ΦPSII (r = 0.93∗∗∗), Rfd (r = 092∗∗∗), and E (r = 091∗∗∗). Chlorophyll content decreased as the photosynthetic efficiency and chlorophyll fluorescence parameters decreased. Hypotheses 1 and 2 were verified in the results of this study. In previous studies, a decrease in chlorophyll content deteriorated the photochemical process, and the dependence of light absorption and fluorescence emission on the concentration of chlorophyll molecules in chloroplasts was demonstrated (Nyachiro et al., 2001). In the present study, proline content negatively correlated with all variables except for WUE (0.69*). Proline content increased as PIABS (−0.78**), E (−0.75**), and Pn MAX (−0.740**) decreased. Proline accumulation is believed to play an adaptive role in plant stress tolerance (Verbruggen & Hermans, 2008). Proline accumulation has been used as a selection parameter for stress tolerance (Yancey et al., 1982; Jaleel et al., 2007). In this study, Pn MAX, E, and PIABS were able to confirm drought stress level at an early stage through a significant correlation with proline accumulation (Fig. 9).

Figure 9 Schematic diagram of the changes of major parameters during progressive treatment time under drought stress condition.

Conclusion

After 25 days of drought stress treatment, the fresh weight of Sargent’s cherry trees decreased by 20.5% compared to the control trees. Photosynthetic efficiency was affected after 15 days of drought stress treatment. When the soil moisture content fell below 10.0%, the decrease in Pn MAX, E, and gs was striking, and WUE temporarily increased. The chlorophyll fluorescence analysis showed that in the early stage of drought stress, energy absorbed per leaf area and energy captured by the photochemical process decreased. ΦPSII, Rfd, NPQ, and PIABS were all positively correlated with photosynthetic efficiency, chlorophyll content, and proline content and were suitable indicators for confirming the level of drought stress. When the soil moisture content fell below 10%, Sargent’s cherry trees avoided hydraulic failure by maintaining water potential through stomatal conductance reduction. These trees were able to temporarily increase water utilization efficiency to reduce water loss inside the leaves while maintaining photosynthetic efficiency. As the soil moisture content dropped below 10.0%, the drought stress response of Sargent’s cherry trees reached its limit, and the loss of electrons in the process of transferring electrons from photosystem II to photosystem I increased, resulting in a significant drop in overall photosynthetic activity. Chlorophyll content also decreased. As the soil moisture content fell below 5.0%, the Pn MAX, E, gs, and chlorophyll fluorescence parameters decreased significantly, and the proline content increased, leading to permanent damage and plant death. Therefore, maintaining soil moisture content above 5% is necessary for the healthy growth of 4-year-old Sargent’s cherry trees. This study identified early physiological indicators that can be used to diagnose and manage the damage caused by drought stress in Sargent’s cherry trees. These results can be used to select the species of other woody plants that are best able to cope with climate change.

Supplemental Information

Supplemental Information 1 Data

Click here for additional data file.

Additional Information and Declarations

Competing Interests

Author Contributions

Data Availability

The authors declare there are no competing interests.

Eon Ju Jin conceived and designed the experiments, performed the experiments, analyzed the data, prepared figures and/or tables, and approved the final draft.

Jun-Hyuk Yoon conceived and designed the experiments, authored or reviewed drafts of the article, and approved the final draft.

Hyeok Lee performed the experiments, analyzed the data, prepared figures and/or tables, and approved the final draft.

Eun Ji Bae conceived and designed the experiments, authored or reviewed drafts of the article, and approved the final draft.

Seong Hyeon Yong performed the experiments, prepared figures and/or tables, and approved the final draft.

Myung Suk Choi conceived and designed the experiments, authored or reviewed drafts of the article, and approved the final draft.

The following information was supplied regarding data availability:

The raw data is available as a Supplemental Files.

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
