# Peer review of "Evaluation of drought stress level in Sargent’s cherry (Prunus sargentii Rehder) using photosynthesis and chlorophyll fluorescence parameters and proline content analysis"

_PeerJ, doi:10.7717/peerj.15954_

## Round 0.1 · original submission · Major Revisions

Eon Ju Jin,

We received two evaluations of your paper. Although one of the reviewers suggests rejection, I allow you to revise your paper.

The main problem the mentioned reviewer finds concerns the low number of replicates. At least, mention this weakness in the manuscript to avoid a similar experimental setup being replicated in future studies. I also find a basic statistical problem in the abstract, where it is written that p was lower than zero, which is clearly impossible. I also confirm the problems with the English grammar: please submit a revised manuscript only after careful professional editing of the language.

Sincerely,

Leonardo Montagnani

Reviewer 1 ·

Basic reporting

Dear Authors,
This manuscript describes time-course changes in several physiological parameters of Prunus sargentii Rehder exposed to drought stress. I believe that this topic is very interesting, as it try to explain physiological aspects of susceptibility to drought in the studied species, especially that it is used for urban or commercial plantings. However, several aspects make this article hard to follow. The most weighty comment is associated with method for measurement of chlorophyll a fluorescence. Additionally, language makes this article rather hard to follow. Moreover, the ‘Discussion’ is almost descriptive, i.e., it is very hard to find any proposed explanations of the observed effects as well as their implications for practice. This is why I recommend to reject this manuscript. However, I believe that this topic and article should not be abandoned. It can be improved, but the amount of work that needs to be done strongly exceeds major revision.

Experimental design

The experimental setup is fine with one exception: the method for measurement of chlorophyll a fluorescence. This needs to be fixed.

Validity of the findings

Results are presented correctly. However, small number of repetitions and ANOVA testing without any statement about data normality make hard to find if the data met the assumption for ANOVA. Additionally, the 'Discussion' section is almost purely descriptive.

Additional comments

MAJOR ISSUES

Language in some fragments is relatively poor (for example L58-68, L88-89, L328, L334-335, etc.) which makes this manuscript very hard to follow. I recommend to check language throughout the manuscript.

The ‘Discussion’ is almost purely descriptive. It is difficult to find an in-depth explanation of the observed results.

Unfortunately, I am not able to see how many plants per treatment per time point were used for each analysis. This needs to be strictly described. How many plants in total were used? How many plants per variant were used? How many plants per time point were used?

For each measured parameter, present and describe in detail unit of measurement when you describe method.

A detailed list of the tested hypotheses is missing. Please state 2-4 hypotheses related to your study, e.g., related to resistance of the studied species to drought, magnitude of changes upon stress, etc.

Main conclusion is missing. In my opinion, the presented data clearly showed that the studied plant species failed to employ any resistance mechanism, which was reflected in all the measured parameters. This is, however, not presented in the presented manuscript; thus, the audience unfortunately does not know if it is worth planting this tree more often or not.

The major problem in this article is the sampling of chlorophyll fluorescence. You stated that chlorophyll fluorescence measurements were conducted according to the study of Yoo et al., 2012 but this author used different fluorometer, namely, Fluor Cam (visualization of chlorophyll fluorescence; produced by PSI from Czech Republic) and you used handheld-type device (OSI 30P; please note that this device is produced by Opti-Sciences, Hudson, NH, USA). Additionally, I believe that for many works on chlorophyll a fluorescence, it is better to state that the calculations were conducted according to the preprogrammed equations of the device. They are mostly or solely based on the works of Stasser et al. (2004). This is, however, a minor problem. Unfortunately, you decided to measure chlorophyll a fluorescence-related parameters only in three repetitions per variant. It is rather uncommon practice with handhelds, as they acquire data from small areas. As there is high heterogeneity of leaf lamina in terms of chlorophyll fluorescence (Lichtenthaler et al., 2005; Murchie and Lawson, 2013), it is reasonable to conduct a greater number of technical measurements (in most recent studies, 2, 4 or 8 per plant), even when you have three true biological replicates. A greater number of measurements allows a more accurate estimation of chlorophyll a fluorescence being close to the average value per leaf. Please give a rationale for number of measurements you conducted. I understand that measurements of NPQ are more time consuming than OJIP protocol. This is, however, worth doing. As I am not very familiar with the OSI 30P device, I tried to determine if the time of measurement was an obstacle. I noticed that the manufacturer did not list NPQ measurements in the device brochure. Was NPQ measured with OSI 30P device or was is calculated from measured values? If calculated, how about protocol for measurements of dark- and light-adapted state? This requires more detailed information.

L47: When you mark time period, please use long dash.
L59: This sentence needs to be fixed.
L69-72: This fragment should be removed as it is too general.
L152-156: Please separate description of soil measurements and plant fresh weight measurements. Plant weight measurements are not clear – please rewrite this.
L163: How many plants per treatments were measured?
L195-197: These formulas are not needed to be presented as you already cite source for calculation.
L213-217: How data normality was tested? Did data meet assumptions for ANOVA testing?
L321-323: You studied effects of relatively short (1 month) stress on plant growth. Thus, I believe that most changes resulted from changes in turgor (wilting) and eventually leaf abscission. It is rather not common to observe clearly marked changes in the growth of woody species during one month of stress.
L343: Always report p < 0.001 but not 0.000.

Lichtenthaler, H.K., Langsdorf, G., Lenk, S., Buschmann, C., 2005. Chlorophyll fluorescence imaging of photosynthetic activity with the flash-lamp fluorescence imaging system. Photosynthetica 43, 355–369. https://doi.org/10.1007/s11099-005-0060-8
Murchie, E.H., Lawson, T., 2013. Chlorophyll fluorescence analysis: A guide to good practice and understanding some new applications. J. Exp. Bot. 64, 3983–3998. https://doi.org/10.1093/jxb/ert208
Strasser, R.J., Tsimilli-Michael, M., Srivastava, A., 2004. Analysis of the chlorophyll a fluorescence transient. In: Papageorgiou, G.C., Govindjee (Eds.), Chlorophyll a Fluorescence. Advances in Photosynthesis and Respiration, vol. 19. Springer, Dordrecht, pp. 321–362. https://doi.org/10.1007/978-1-4020-3218-9_12
Yoo, S.Y., Eom, K.C., Park, S.H., Kim, T.W., 2012. Possibility of drought stress indexing by chlorophyll fluorescence imaging technique in red pepper (Capsicum annuum L.). Korean Journal of Soil Science and Fertilizer 45: 676-682 https://doi.org/10.7745/KJSSF.2012.45.5.676

Reviewer 2 ·

Basic reporting

It is written in English, and the object is clear.
Literature references and sufficient field background are provided.
Tables and figures are professionally structured.
There was no hypothesis in this manuscript.

Experimental design

The research is within the Aims and Scope of the journal.
The research was conducted well, and it was not mentioned that the study contributes to filling the knowledge gap.
The investigation was conducted rigorously and to a high technical standard.
Methods were described with sufficient information.

Validity of the findings

The impact and novelty were not found in this study; however, the results are generally acceptable and prove that knowledge is validated.
The conclusions were appropriately stated.

Additional comments

The study was well conducted with controlled experiments. Parameters were also appropriately chosen to measure the stress.
References are well cited to support the results.
The study's finding was not new in the field; however, it is relevant and acceptable to the audience.

---

## Round 0.2 · Major Revisions

Dear Dr. Yong and Dr. Choi,

We received one evaluation for the revised version of your manuscript.

In addition to the reviewer's comments, please consider carefully the methodological part of the study, in order to make it fully reproducible.

Sincerely,

Leonardo Montagnani

Reviewer 2 ·

Basic reporting

It is very common that the physiological responses to the drought stress were measured in the parameters shown in this manuscript.
It contains valuable data on the physiological measurement for the plant physiologists. However, so it had no hypothesis although the results are relevant.
It needs English improvement in explanations.

Experimental design

See the comments below;

line 183 Analysis of chlorophyll fluorescence
line 185~188, leaf samples are probably different age and chlorophyll fluorescence is influence by the age.

line 200 Analysis of chlorophyll contents
line 209 Analysis of proline contents
In both experiments, MM stated samples were collected one time 30 days after the treatment, however there were measurement data every 5 days. It must be clear.

line 222 Statistical analysis
Line 229, It is not clear which data were used in Pearson's correlation analysis, because there are many data in each parameter.

The experiment has not clear objectives and some methodology does not sound.
Research question is not well defined, Similar results were published by many researchers and the results of this manuscript did not fill an identified knowledge gap.

Validity of the findings

Conclusions are well stated.

---

## Round 0.3 · Minor Revisions

Dear Dr. Yong and Dr. Choi,

We carefully evaluated your paper. We still have found some details that have to be fixed before publication, In particular, the P-value should be reported either in its numeric form or below a given threshold, generally <0.05. In addition, I recommend that your paper is edited by a professional editor since some sentences are still unclear.

Sincerely,

Leonardo Montagnani

---

## Round 0.4 · accepted · Accept

Dear Dr. Yong and Dr. Choi,

I am pleased to inform you that I consider your paper acceptable now.

Congratulations!

Sincerely,

Leonardo Montagnani